# Universal Metric Learning with Parameter-Efficient Transfer Learning

## Abstract

A common practice in metric learning is to train and test an embedding model for each dataset. This dataset-specific approach fails to simulate real-world scenarios that involve multiple heterogeneous distributions of data. In this regard, we introduce a novel metric learning paradigm, called Universal Metric Learning (UML), which learns a unified distance metric capable of capturing relations across multiple data distributions. UML presents new challenges, such as imbalanced data distribution and bias towards dominant distributions. To address these challenges, we propose Parameter-efficient Universal Metric leArning (PUMA), which consists of a pre-trained frozen model and two additional modules, stochastic adapter and prompt pool. These modules enable to capture dataset-specific knowledge while avoiding bias towards dominant distributions. Additionally, we compile a new universal metric learning benchmark with a total of 8 different datasets. PUMA outperforms the state-of-the-art dataset-specific models while using about 69 times fewer trainable parameters.

## 1 Introduction

Learning semantic distance metrics has been playing a key role in machine learning applications including content-based image retrieval (Kim et al., 2019; Movshovitz-Attias et al., 2017; Sohn, 2016; Song et al., 2016), face verification (Liu et al., 2017; Schroff et al., 2015), person re-ID (Chen et al., 2017; Xiao et al., 2017), few-shot learning (Qiao et al., 2019; Snell et al., 2017; Sung et al., 2018), and representation learning (Kim et al., 2019; Wang & Gupta, 2015; Zagoruyko & Komodakis, 2015). Deep metric learning has served as a representative approach to learn semantic distance metrics: it aims to learn highly nonlinear distance metrics through deep neural networks that approximate the actual underlying semantic similarity between samples.

While metric learning methods have achieved remarkable progress, they focus only on learning metrics for a specific domain. However, real-world applications often violate this assumption and involve multiple heterogeneous data distributions. For instance, users of a retrieval system may query data of substantially different semantics that form multiple diverse distributions. To tackle this issue using conventional methods, it is imperative to train multiple models as shown Fig. 1(a) and subsequently combine them through ensemble techniques or toggle between the models based on the query. Such procedures are not only arduous but also demand a significant amount of computational resources.

In this paper, we introduce a new metric learning paradigm, called **Universal Metric Learning (UML)**. UML aims to learn a unified distance metric capable of capturing semantic similarity across multiple data distributions. Instead of learning a model per dataset, UML trains a single model on the union of multiple heterogeneous datasets to create a universal embedding space.

UML opens a new fascinating direction towards metric learning in the wild, but at the same time comes with technical challenges not found in the conventional metric learning. First, integrating multiple datasets results in highly imbalanced data distributions. Data imbalance is a natural phenomenon and is a well-known challenge in many recognition tasks, as it induces bias and hinders performance. In addition to common imbalance issues such as class imbalance tackled by recent work (Liu et al., 2019; Zhong et al., 2021), UML introduces a more complex and unique challenge caused by *dataset imbalance* when datasets to be integrated are of substantially different sizes. Our study reveals that a naïve fine-tuning on multiple datasets as a whole, depicted in Fig. 1(b), re-

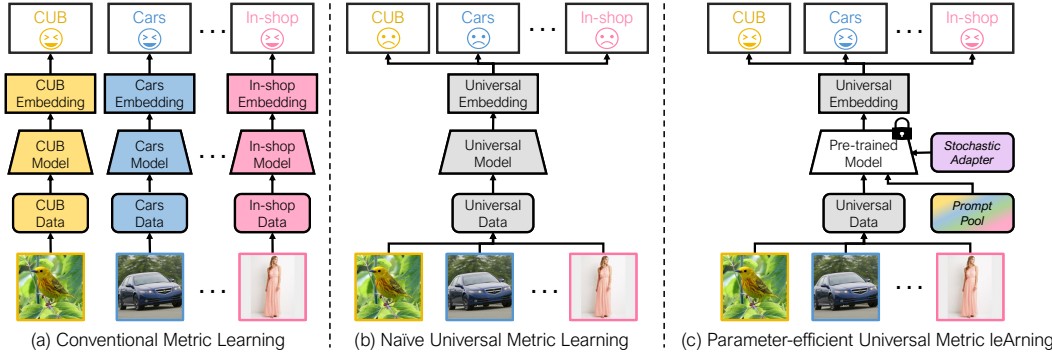

(a) Conventional Metric Learning  (b) Naïve Universal Metric Learning  (c) Parameter-efficient Universal Metric leArning

Figure 1: **Comparison between conventional and universal metric learning methods.** (a) Conventional metric learning employs separate models for individual datasets, incurring significant computational and memory costs as data diversity grows. (b) A naïve solution is to fine-tune the model on a merged dataset, but this often leads to a severe bias towards major data distributions. (c) In contrast, our method excels on all datasets with just one model. This is highly resource-efficient as it enables one-time learning and evaluation on diverse data distributions using a single model.

sults in models strongly biased towards large datasets. Second, key features for discriminating between classes may vary across datasets. For example, color is useful for differentiating between bird species but harmful for distinguishing vehicle types. Thus, models should learn to recognize both dataset-specific discriminative features and common discriminative features to achieve UML.

To address these challenges, we propose a novel approach called **Parameter-efficient Universal Metric leArning (PUMA)**, which is a completely different direction from existing metric learning. PUMA aims to capture universal semantic similarity through a single embedding model while mitigating the imbalance issues. To achieve this, we draw inspiration from recent advances in parameter-efficient transfer learning in natural language processing (Houlsby et al., 2019; Pfeiffer et al., 2020; He et al., 2021; Li & Liang, 2021). Our key idea is to freeze the parameters of a model pre-trained on a large-scale dataset, thereby preserving its generalization capability, and learn dataset-specific knowledge from the unified dataset with a minimal number of additional parameters.

Specifically, PUMA is built on a pre-trained Vision Transformer (ViT) and incorporates two additional modules, namely *stochastic adapters* and a *prompt pool* (see Fig. 1(c)). A stochastic adapter is a lightweight module that operates in parallel with the corresponding transformer block, and its operation is stochastically switched off during training. It enables the pre-trained model to adapt, while avoiding being biased towards a specific data distribution by randomly providing either adapted features or features from the pre-trained model. It is parameter-efficient yet effective, and improves performance across all datasets without bias. Meanwhile, the prompt pool is used to build a conditional prompt that accounts for distinct characteristics of each dataset on the fly. To be specific, the prompt pool is a set of prompts organized in a key-value memory, and given the input feature, the conditional prompt is generated by aggregating relevant prompts in the pool using an attention mechanism. The conditional prompt is added to the input sequence of the ViT, allowing for more targeted adaptation.

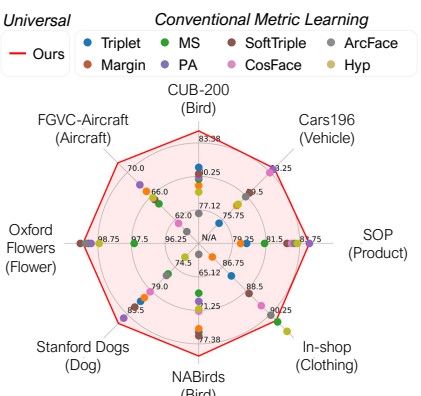

Figure 2: Our single model trained on the 8 datasets outperformed most existing models devoted to each dataset while using about 69 times fewer trainable parameters.

We compile a new universal metric learning benchmark with a total of 8 datasets of different domains and classes. Our method largely outperformed models trained on multiple datasets using conventional metric learning techniques, and even surpassed most of the models trained on each dataset (*i.e.*, dataset-specific models) with about 69 times fewer trainable parameters as shown in Fig. 2. In addition, we demonstrate that our method also can be utilized as a strong few-shot learner.

## 2 RELATED WORK

**Deep Metric Learning.** It aims to learn a metric function that approximates the underlying semantic similarity of data by pulling semantically similar samples (positive) closer to the anchor and pushing dissimilar samples (negative) away. To achieve this goal, the development of loss functions has been the main focus of this field, typically classified into pair-based and proxy-based losses. Pair-based losses consider relations between pairs (Wu et al., 2017; Bromley et al., 1994; Chopra et al., 2005; Hadsell et al., 2006), triplets (Wang et al., 2014; Schroff et al., 2015) or higher-order tuples of samples (Song et al., 2016; Sohn, 2016; Wang et al., 2019a;b; Song et al., 2017). They can capture the fine-grained relations among samples, but they are suffering from the issue of increased training complexity as the number of training data increases. Proxy-based losses address the complexity issue by introducing learnable parameters called proxies to represent the training data of the same class. They greatly reduce the complexity of examining the relations between all data by considering those between data and proxies. In this direction, the approaches have used proxies to approximate the pair-based loss (Movshovitz-Attias et al., 2017; Kim et al., 2020; Qian et al., 2019) or have modified the cross-entropy loss (Deng et al., 2019; Wang et al., 2018; Zhai & Wu, 2018; Teh et al., 2020). Although there have been remarkable advances in metric learning so far, all existing metric learning methods deal only with a specific distribution within one dataset. Orthogonal to them, we first shed light on the new problem of learning a *universal metric* contained in multiple distributions and explore ways to address it. While prior work mostly focuses on the design of loss functions, we also explore the impact of architectural choices in metric learning.

**Parameter-efficient Transfer Learning.** Large-scale pre-trained models have shown significant improvements across various downstream tasks. As the model size and the number of tasks grow, parameter-efficient transfer learning approaches (Hu et al., 2021; Rebuffi et al., 2017; Houlsby et al., 2019; Pfeiffer et al., 2020; He et al., 2021) have been developed to adapt to diverse downstream by updating only a small fraction/number of learnable parameters while fully utilizing the knowledge of the pre-trained model without catastrophic forgetting. Especially in the field NLP, low-rank adaptation (Hu et al., 2021) is proposed to approximate the parameter update or light-weight adapter modules (Houlsby et al., 2019; Pfeiffer et al., 2020) can be inserted between pre-trained layers during fine-tuning. Prefix/prompt tuning (Lester et al., 2021; Li & Liang, 2021; Wang et al., 2022; Smith et al., 2022) has been introduced where additional learnable tokens (soft prompts) are added during fine-tuning while keeping the backbone frozen. In contrast to prior deep metric learning work, we utilize parameter-efficient transfer learning for our proposed universal metric learning setup to learn universal representations across different data distributions in a single model while preventing bias and catastrophic forgetting, which outperforms even full fine-tuning baselines.

## 3 UNIVERSAL METRIC LEARNING

In this section, we first review conventional metric learning, and then introduce the UML setting and discuss its technical challenges.

### 3.1 REVISITING CONVENTIONAL METRIC LEARNING

Metric learning is the task of learning a distance function that captures the semantic dissimilarity between samples in a given dataset $S$. Such a distance function $d$ holds:

$$d(x, x^+; \theta) < d(x, x^-; \theta) \quad \forall (x, x^+, x^-), \tag{1}$$

where $x^+$ and $x^-$ denote the positive sample that belongs to the same class as $x$, and negative samples that are not, respectively, and $\theta$ represents the model parameters. Deep metric learning achieves this by learning a deep neural network as a high-dimensional embedding function, and employing Euclidean or cosine distance to calculate the distance between embedding vectors.

Note that metric learning seeks a generalization to classes unseen in training. The conventional setup thus employs a set of classes $C_t$ and their labeled samples $S_t = \{(x_t, y_t) \mid y_t \in C_t\}$ for training, and evaluates a trained embedding model for a set of unseen classes $S_u = \{(x_u, y_u) \mid y_u \in C_u\}$, where $C_t \cap C_u = \varnothing$ and $S_t \cup S_u = S$. This convention only considers generalization within a single dataset.

## 3.2 PROBLEM FORMULATION OF UML

UML is an extension of the conventional one and tackles the challenging and practical problem of dealing with **multiple datasets drawn from different data distributions**, using a single embedding model. The goal of UML is to learn a universal distance metric that can effectively capture diverse relations among samples across multiple datasets, while maintaining the intra-class compactness and inter-class separability within each dataset. In UML, a model is trained *as if it were given a single dataset, without knowing that multiple datasets are combined*, making it highly suitable for both a large-scale dataset drawn from a multimodal distribution and a union of multiple small datasets in real-world applications.

Suppose that we have $N_s$ datasets, denoted as $S^1, S^2, \cdots, S^{N_s}$, and define the unified dataset $\mathbb{S} = \cup_{i=1}^{N_s} S^i$. To learn a universal embedding function, UML leverages the unified training dataset $\mathbb{S}_t = \cup_{i=1}^{N_s} S_t^i$, which aggregates training data from all the datasets. The learned universal distance function is evaluated in two different ways to assess its generalization capability. First, we evaluate it on the unified unseen test data $\mathbb{S}_u = \cup_{i=1}^{N_s} S_u^i$ to measure its universal accuracy, which demonstrates its capacity to comprehend semantic similarity across all datasets without favoring any specific dataset. Second, we evaluate the distance function on the unseen test data of each dataset $S_u^i$ separately, to assess its ability to grasp the specific semantic similarity for each dataset.

## 3.3 CHALLENGES IN UML

UML encounters a new challenge – *highly imbalanced distribution of the unified dataset*. Data imbalance is a common and well-known issue in a large variety of vision tasks. However, UML presents a more complex and unique challenge, where the entire data distribution becomes long-tailed due to class imbalance, and also has dataset imbalance caused by integrating datasets of substantially different sizes, as illustrated in Fig. 5. This issue is particularly critical in metric learning: the dataset imbalance results in a substantial portion of samples within the batch originating from larger datasets, and consequently, the model tends to focus predominantly on learning relations within these larger datasets, thereby introducing a dataset bias.

Another challenge in UML is that *class-discriminative features are not shared across all datasets.* This challenge arises due to the disparity between different data distributions since each distribution has its own characteristics that define relations between its samples, which could conflict with those of the other distributions. For instance, while color may be crucial for differentiating between bird species, it may impede distinguishing between different vehicle types. Thus, training with a unified dataset may lead to two potential problems. First, if the model focuses on class-discriminative features that are specific to a certain data distribution, it may have a negative impact on datasets where those features are not relevant. Second, if the model attends to the commonalities shared among all datasets, its discriminability for capturing fine-grained differences between samples may diminish.

Moreover, UML still has a challenge in *generalization to unseen classes*, inherited from conventional metric learning. However, this challenge becomes even more difficult as UML deals with diverse imbalanced distributions.

Adopting the traditional strategy of training multiple models and subsequently ensembling them is a straightforward way to handle diverse datasets without confronting the above challenges. However, this approach demands a vast number of parameters and substantial computational resources. Instead, we will introduce a parameter-efficient approach that elegantly tackles all the aforementioned challenges.

## 4 PROPOSED METHOD

We propose a novel approach to UML, named Parameter-efficient Universal Metric leArning (PUMA). In contrast to conventional metric learning methods that fine-tune the entire model parameters, PUMA does not tune a large-scale pre-trained model but keeps its generalization capability across diverse data distributions. Instead, we leverage small additional modules that learn dataset-specific knowledge from the unified dataset. As shown in Fig. 3, PUMA uses a pre-trained ViT as a backbone, and employs stochastic adapters and a prompt pool as the additional modules, which are detailed in the remainder of this section.

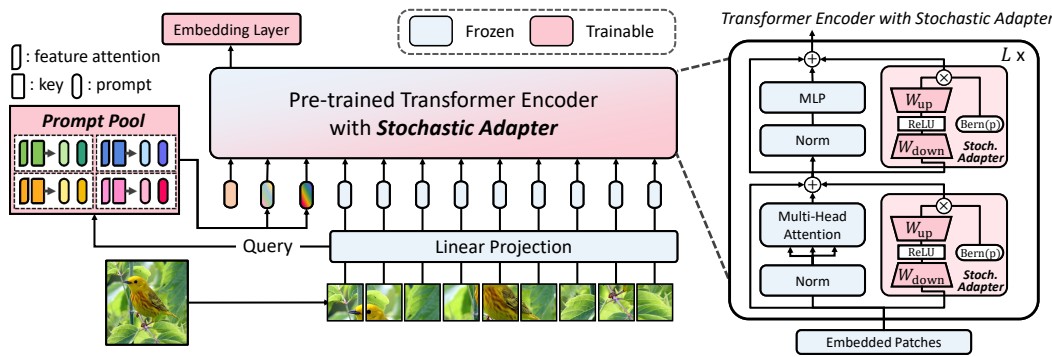

Figure 3: **An overview of PUMA.** PUMA consists of two learnable modules: stochastic adapters (Sec. 4.2) and a prompt pool (Sec. 4.3). Using the output of the transformer's embedding layer as a query, and it creates a conditional prompt by integrating relevant prompts through an attention mechanism. The conditional prompt is combined with image embeddings and class token, and then fed into the transformer. The modified input is embedded through transformer blocks, each coupled with a stochastic adapter, a learnable bottleneck module that turns on stochastically during training.

## 4.1 PRELIMINARIES: VIT

ViT (Dosovitskiy et al., 2021) is composed of a patch embedding layer and an encoder with $L$ sequential transformer layers. The patch embedding layer splits the input image $x$ into image patch embeddings $E \in \mathbb{R}^{N_e \times D}$, where $N_e$ denotes the number of patch embeddings and $D$ is the embedding dimension. The input sequence of the transformer encoder is formed by appending the image patch embeddings to a learnable class token embedding $e_{\text{cls}} \in \mathbb{R}^D$, as follows:

$$z_0 = [e_{\text{cls}}, E]. \tag{2}$$

Each transformer layer consists of multi-headed self-attention (MSA) and multilayer perceptron (MLP) blocks, with layer normalization (LN) applied before every block and residual connections after every block:

$$\begin{aligned} z'_\ell &= \text{MSA}(\text{LN}(z_{\ell-1})) + z_{\ell-1}, & \ell = 1, \ldots, L, \\ z_\ell &= \text{MLP}(\text{LN}(z'_\ell)) + z'_\ell, & \ell = 1, \ldots, L, \end{aligned} \tag{3}$$

## 4.2 STOCHASTIC ADAPTER

To effectively adapt the model to the unified dataset without being biased to the large dataset, we propose a stochastic adapter. While adding learnable parameters shared by all data enables adaptation, this could cause the additional parameters to be biased towards the major distribution due to the imbalanced distribution issue. We resolve this issue by stochastic adaptation, which allows an embedding space to consider both the generalizable features of a pre-trained model and adapted features, rather than relying solely on the adapted features. This alleviates bias in the embedding space toward the major data distribution, while providing the capacity to learn knowledge specific to each dataset. Our adapter has a bottleneck structure for parameter-efficiency and is connected in parallel with every transformer block. The adapter consists of a down-projection layer $W_{\text{down}} \in \mathbb{R}^{D \times r}$, a ReLU activation layer, and an up-projection layer $W_{\text{up}} \in \mathbb{R}^{r \times D}$, where $r < D$ is the bottleneck dimension. As shown in Fig. 3, within a transformer layer, two adapters are placed in parallel, one with the MSA block and the other with the MLP block. Given input for the $\ell$-th transformer layer and output of the $\ell$-th MSA layer, outputs of the adapters are produced as follows:

$$\begin{aligned} \tilde{z}'_\ell &= \text{ReLU}(\text{LN}(z_{\ell-1}) \cdot W'_{\text{down}}) \cdot W'_{\text{up}} \\ \tilde{z}_\ell &= \text{ReLU}(\text{LN}(z'_\ell) \cdot W_{\text{down}}) \cdot W_{\text{up}}, \end{aligned} \tag{4}$$

where $W'_{\text{down}}$ and $W'_{\text{up}}$ have the same shapes as $W_{\text{down}}$ and $W_{\text{up}}$, respectively. The output features of the adapters are multiplied by random binary masks and combined with the outputs of the transformer blocks (*i.e.*, MSA and MLP) through residual connections:

$$\begin{aligned} z'_\ell &= \text{MSA}(\text{LN}(z_{\ell-1})) + z_{\ell-1} + \gamma'_\ell \cdot \tilde{z}'_\ell, \\ z_\ell &= \text{MLP}(\text{LN}(z'_\ell)) + z'_\ell + \gamma_\ell \cdot \tilde{z}_\ell, \end{aligned} \tag{5}$$

where $\gamma'_\ell$ and $\gamma_\ell$ are independent variables drawn from Bernoulli($p$), and $p$ is the keep probability of the stochastic adapters.

### 4.3 CONDITIONAL PROMPT LEARNING

We propose conditional prompt learning[1] to learn more discriminative features for each dataset. We assume that images within each dataset exhibit shared characteristics distinguished from those of other datasets. Our goal is to learn and leverage prompts relevant to the input data among the set of prompts through the attention mechanism. To achieve this, a query feature that encodes the input image $x$ is first extracted. Query features should be able to grasp the data distribution of the input image and also require little computation to obtain it. Considering these requirements, we design a simple query feature for $x$ by using a pooling operation on its image patch embeddings $E$ in Sec. 4.1:

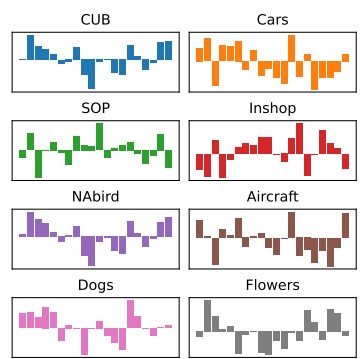

Figure 4: The average similarity between input queries and prompts for each dataset. The $x$-axis represents prompt index.

$$q = \text{AvgPool}(E) + \text{MaxPool}(E), \quad q \in \mathbb{R}^D. \tag{6}$$

Then, we introduce a *prompt pool*, a storage that contains prompts together with extra parameters for input-conditioning. $P_m \in \mathbb{R}^{N_p \times D}$ denote a prompt in the pool, where $N_p$ is token length of a prompt, and then a prompt pool with $M$ prompts is given by:

$$\mathbf{P} = \{(P_1, K_1, A_1), \cdots, (P_M, K_M, A_M)\}, \tag{7}$$

where $K_m \in \mathbb{R}^D$ denotes the key of a prompt, and $A_m \in \mathbb{R}^D$ is its feature attention vector, a learnable parameter emphasize specific feature dimensions of the query vector. The query feature is element-wise multiplied with the feature attention vector to create an attended query, which is then paired with the prompt key for matching. The weight vector is computed based on the cosine similarity between the attended query and the prompt key, which is given by

$$\alpha_m = s(q \otimes A_m, K_m), \tag{8}$$

where $s(\cdot, \cdot)$ denotes the cosine similarity between two vectors and $\otimes$ denotes element-wise product operation over the feature dimension. The conditional prompt of input image $x$ is calculated as a weighted sum of prompts:

$$\hat{P} = \sum_{m=1}^{M} \alpha_m P_m, \tag{9}$$

Finally, it is inserted into the input sequence of the transformer encoder:

$$z_0 = [e_{\text{cls}}, \hat{P}, E]. \tag{10}$$

This process allows each prompt to condition images based on their specific data distributions, as depicted in Fig. 4. Notably, while the CUB dataset demonstrates a strong tendency to align with the relevant NABird dataset, it distinctly prefers different prompts compared to the In-shop dataset.

## 5 EXPERIMENTS

### 5.1 EXPERIMENTAL SETUP

**Datasets.** In the UML setting, we employ a combination of eight datasets. These comprise four widely recognized benchmarks: CUB (Welinder et al., 2010), Cars-196 (Krause et al., 2013), Standford Online Product (SOP) (Song et al., 2016), and In-shop Clothes Retrieval (In-Shop) (Liu et al., 2016). Alongside these, we incorporate other four fine-grained datasets: NABirds (Van Horn et al., 2015), Dogs (Khosla et al., 2011), Flowers (Nilsback & Zisserman, 2008), and Aircraft (Maji et al., 2013). The overall dataset statistics are in Table 1. The combined dataset encompasses 141,404 training images and 148,595 testing images. Notably, this dataset exhibits imbalanced data distributions, with a significant portion of images from large-scale datasets such as SOP and In-Shop, as shown in Fig. 5. We also provide results on the conventional benchmarks in Appendix D.1.

**Baselines.** We benchmark our method against three distinct learning strategies. The models trained exclusively on individual datasets are termed **dataset-specific models**, while models trained on multiple datasets are termed **universal models**.

---

[1]The prompt in ViT denotes the learnable input token parameters as in Jia et al. (2022); Wang et al. (2022); Smith et al. (2022)

Table 1: Dataset statistics: the number of training images and their classes used in training and testing.

|  | CUB | Cars | SOP | In-shop | NABirds | Dogs | Flower | Aircraft | Total |
|---|---|---|---|---|---|---|---|---|---|
| Train Samples | 5.8K | 8.0K | 59.5K | 25.8K | 22.9K | 10.6K | 3.5K | 5K | 141.4K |
| Train Classes | 100 | 98 | 11.3K | 3.9K | 278 | 60 | 51 | 50 | 15.9K |
| Test Samples | 5.9K | 8.1K | 60.5K | 28.7K | 25.6K | 9.9K | 4.7K | 5K | 148.5K |
| Test Classes | 100 | 98 | 11.3K | 3.9K | 277 | 60 | 51 | 50 | 15.9K |

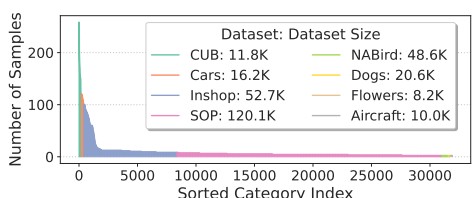

Figure 5: The number of samples in each category. Each color represents its dataset.

(a) **Dataset-specific models by full fine-tuning**: These models employ conventional metric learning protocols, where every parameter in the backbone and the embedding layer is fully updated. For this approach, we utilize a range of renowned metric learning loss functions, Triplet (Schroff et al., 2015), Margin (Wu et al., 2017), MS (Wang et al., 2019b), Proxy-Anchor (PA) (Kim et al., 2020), SoftTriple (Qian et al., 2019), CosFace (Wang et al., 2018), ArcFace (Deng et al., 2019), CurricularFace (Huang et al., 2020), and Hyp (Ermolov et al., 2022). Notably, each model is trained specifically for individual datasets.

(b) **Universal models by full fine-tuning**: The models are fully fine-tuned using the aforementioned loss functions, leveraging a union of multiple datasets.

(c) **Universal models by parameter-efficient fine-tuning**: The models update a subset of backbone parameters or add new trainable parameters to the backbone during the fine-tuning. We explore two techniques focusing on the embedding layer: training solely the linear embedding layer (Linear Emb.) and the embedding layer enriched with a 3-layered multilayer perceptron (MLP-3 Emb.). Further, we consider three prominent parameter-efficient tuning strategies: VPT (Jia et al., 2022), LoRA (Hu et al., 2021) and AdaptFormer (Chen et al., 2022). LoRA and AdaptFormer are scaled with the same parameters as our method.

**Implementation Details.** For fair comparisons, all models are evaluated using the same backbone, ViT-S/16 (Dosovitskiy et al., 2021) pre-trained on ImageNet-21K (Deng et al., 2009) following previous work (Ermolov et al., 2022). We change the size of its last linear layer to 128, and $L_2$-normalize the output embedding vector. We set the parameters for the stochastic adapter to $r = 128$ and $p = 0.5$, for the conditional prompt to $N_p = 8$ and $M = 20$. Unless otherwise specified, we adopt the CurricularFace loss (Huang et al., 2020) as a metric learning loss for parameter-efficient fine-tuning methods, including our method. We also ablate different loss functions on our method in Appendix B.1. More implementation details can be found in Appendix A.

**Evaluation Protocol.** We measure the performance using Recall@1 in the main paper, with additional results with R@$k$, MAP@R, and RP in Appendix D.2. We report the **dataset-specific accuracy** using individual query and gallery for each dataset, and also calculate two kinds of **universal accuracy**, the unified accuracy using unified query and gallery sets and the harmonic mean of these individual accuracies. To evaluate the unified performance of dataset-specific models, we use an ensemble approach, averaging the embedding vectors of all dataset-specific models.

## 5.2 RESULTS

Table 2 shows Recall@1 performance with a total of eight datasets. We note that the total number of parameters of dataset-specific models increases as the number of datasets increases. **(1)** Our results show that PUMA surpasses all compared data-specific models (Table 2(a)) in terms of universal accuracy. Moreover, our method outperforms data-specific models in all cases except for In-Shop and Dog, while not using hyperparameters selected for each respective dataset. Surprisingly, our method accomplishes this level of performance while *using up to 69 times fewer trainable parameters* than previous techniques. This indicates that PUMA can be trained with limited resources and can easily be scaled up to a larger model and more datasets. Furthermore, even without emphasizing parameter efficiency, the outcomes highlight that our method can be a promising alternative to existing dataset-specific metric learning approaches. **(2)** While universal models by full fine-tuning (Table 2(b)) suffer significant performance degradation on small datasets, PUMA consistently achieves high performance across all datasets. Consequently, our results show that PUMA surpasses all compared methods both in terms of dataset-specific accuracy and universal accuracy. It enhances the best performance of universal models in the unified accuracy and harmonic mean accuracy by 3.4% and 4.6%, respectively. **(3)** Among various parameter-efficient fine-tuning methods (Table 2(c)), only our approach manages to outperform the majority of full fine-tuned models. Models like Linear

Table 2: Recall@1 of metric learning baselines and ours on the 8 datasets. Their network architecture is ViT-S/16 (Dosovitskiy et al., 2021) with 128 embedding dimensions. We mark in **bold** the best among all scores in the entire table per evaluation metric.

| | Params (M) | Dataset-Specific Accuracy | | | | | | | | Universal Accuracy | |
|---|---|---|---|---|---|---|---|---|---|---|---|
| Methods | Train / Total | CUB | Cars | SOP | InShop | NABird | Dog | Flowers | Aircraft | Unified | Harmonic |
| (a) *Dataset-specific models by full fine-tuning* | | | | | | | | | | | |
| Triplet | 173.7 / 173.7 | 81.1 | 75.2 | 80.2 | 87.4 | 75.2 | 81.0 | 99.1 | 64.7 | 57.6 | 79.4 |
| Margin | 173.7 / 173.7 | 79.4 | 78.0 | 79.8 | 86.0 | 74.6 | 80.3 | 99.0 | 66.8 | 58.1 | 79.6 |
| MS | 173.7 / 173.7 | 80.0 | 83.7 | 81.4 | 90.8 | 68.1 | 75.8 | 97.4 | 64.7 | 61.6 | 78.9 |
| PA | 173.7 / 173.7 | 80.2 | 83.7 | 84.4 | 91.5 | 69.6 | 84.2 | 99.0 | 67.9 | 57.4 | 81.4 |
| SoftTriple | 173.7 / 173.7 | 80.5 | 80.0 | 82.9 | 88.7 | 75.9 | 82.1 | 99.4 | 65.4 | 63.4 | 80.8 |
| CosFace | 173.7 / 173.7 | 78.8 | 83.2 | 83.2 | 89.6 | 71.4 | 79.2 | 99.2 | 61.4 | 61.5 | 79.3 |
| ArcFace | 173.7 / 173.7 | 76.8 | 79.4 | 83.4 | 90.3 | 61.0 | 76.1 | 99.2 | 60.0 | 58.8 | 76.2 |
| CurricularFace | 173.7 / 173.7 | 79.7 | 81.3 | 83.2 | 88.2 | 75.3 | 81.2 | 99.1 | 63.9 | 62.9 | 80.4 |
| Hyp | 173.7 / 173.7 | 78.8 | 78.2 | 83.6 | 91.5 | 71.0 | 72.6 | 98.7 | 65.7 | 15.7 | 78.8 |
| (b) *Universal models by full fine-tuning* | | | | | | | | | | | |
| Triplet | 21.7 / 21.7 | 74.5 | 35.4 | 80.2 | 85.7 | 68.2 | 77.1 | 98.7 | 40.9 | 72.0 | 57.7 |
| Margin | 21.7 / 21.7 | 72.5 | 36.7 | 80.0 | 84.1 | 67.4 | 74.8 | 98.5 | 40.4 | 71.6 | 57.4 |
| MS | 21.7 / 21.7 | 66.3 | 22.9 | 78.9 | 87.2 | 58.6 | 69.8 | 97.3 | 31.5 | 67.8 | 47.3 |
| PA | 21.7 / 21.7 | 77.2 | 73.1 | 83.7 | 91.9 | 71.5 | 78.1 | 96.4 | 62.7 | 77.9 | 71.0 |
| SoftTriple | 21.7 / 21.7 | 78.9 | 77.0 | 81.3 | 88.6 | 73.8 | 79.3 | 99.1 | 64.4 | 77.6 | 72.7 |
| CosFace | 21.7 / 21.7 | 74.2 | 73.5 | 82.5 | 90.0 | 69.7 | 74.1 | 98.7 | 59.7 | 76.6 | 69.6 |
| ArcFace | 21.7 / 21.7 | 70.8 | 25.9 | 63.9 | 58.9 | 64.0 | 70.3 | 97.2 | 31.7 | 59.2 | 47.2 |
| CurricularFace | 21.7 / 21.7 | 78.3 | 77.9 | 82.0 | 89.1 | 73.0 | 79.3 | 99.1 | 65.6 | 77.9 | 79.5 |
| Hyp | 21.7 / 21.7 | 79.2 | 60.6 | 83.5 | 90.9 | 73.6 | 81.9 | 99.1 | 56.3 | 77.7 | 69.4 |
| (c) *Universal models by parameter-efficient fine-tuning* | | | | | | | | | | | |
| Linear Emb. | 0.1 / 21.8 | 82.1 | 49.7 | 70.5 | 65.5 | 77.9 | **86.2** | 99.1 | 47.6 | 69.3 | 68.2 |
| MLP-3 Emb. | 5.3 / 27.0 | 57.5 | 29.7 | 63.1 | 63.2 | 50.6 | 64.5 | 93.6 | 32.8 | 56.5 | 50.3 |
| VPT | 0.1 / 21.8 | 82.8 | 51.0 | 74.7 | 72.9 | 78.3 | 85.5 | 99.2 | 50.8 | 72.3 | 70.8 |
| LoRA | 2.4 / 24.1 | 77.0 | 70.9 | 81.3 | 86.2 | 70.8 | 79.1 | 98.9 | 59.7 | 76.1 | 76.5 |
| AdaptFormer | 2.4 / 24.1 | 77.0 | 77.0 | 83.7 | 90.7 | 72.3 | 78.5 | 99.0 | 63.9 | 78.5 | 79.0 |
| Ours | 2.5 / 24.2 | **83.9** | **84.3** | **84.0** | 89.8 | **79.2** | 84.1 | **99.3** | **72.6** | **81.3** | **84.1** |

Embedding and VPT, which employ fewer learnable parameters, notably underperform on datasets such as Cars, SOP, In-Shop, and Aircraft, where substantial domain-specific knowledge is required. Both AdaptFormer and LoRA, which use a similar number of parameters as our method, similarly show biases towards large datasets akin to full fine-tuning.

**Discussion on Losses in UML.** We observe that loss functions in UML exhibit different behaviors compared to conventional metric learning. This is due to the significant changes in the intra-class similarity and inter-class separability across datasets, making the effect of loss design and its hyperparameters critical. The choice of margin in a loss greatly affects performance, with large margins (*e.g.*, ArcFace) leading to significant drops in performance. In contrast, sophisticated losses such as SoftTriple which uses multiple proxies, and CurricularFace which dynamically considers hard negatives demonstrate superior performance. In addition, pair-based losses tend to underperform in the UML setting, due to the diminished presence of samples from smaller datasets within a batch, limiting the exploration of pairwise relations among those samples. See Appendix C.1 for further discussion.

**Few-shot Metric Learning.** We additionally demonstrate the data-efficient adaptation capability of PUMA, by exploring few-shot learning in Fig. 6. Different from the prior few-shot learning approaches (*e.g.*, (Chen et al., 2019; Jung et al., 2022)), we train models with few-shot labels on the training classes and evaluate them on unseen classes in a zero-shot manner. The results demonstrate that PUMA shows better performance than linear probing, which is considered a strong few-shot learning baseline (Tian et al., 2020). Moreover, even with an increasing number of shots, PUMA outperforms the fine-tuning model in terms of harmonic mean accuracy.

## 5.3 ABLATION STUDY

**Ablation Study on Each Component.** Table 3 shows an extensive ablation study to analyze the effectiveness of each module in PUMA. We observe that employing a conditional prompt outperforms using a single prompt across multiple datasets. This highlights the adaptability inherent to our

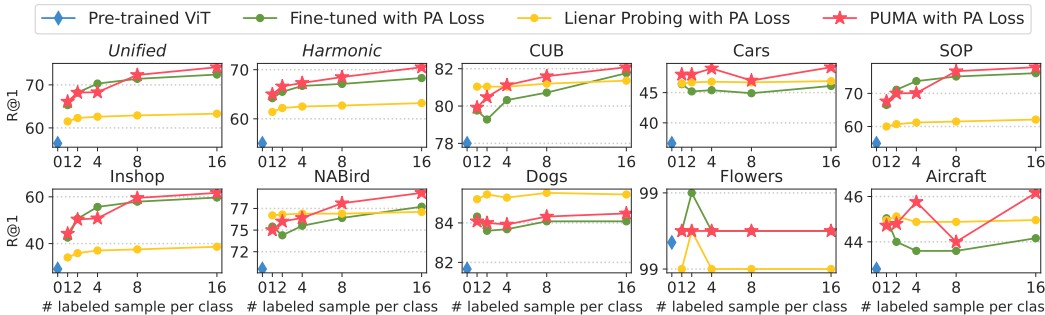

Figure 6: Accuracy in Recall@1 of few-shot metric learning on the 8 datasets. Except for the pre-trained model ViT model, all others are trained with Proxy-Anchor loss.

Table 3: Ablation study on each component of PUMA. "Sing." denotes single prompt ($M = 1$), "Cond." denotes our conditional prompt ($M = 20$), "Stat." denotes adapter with $p = 1$, and "Stoc." denotes our conditional adapter with $p = 0.5$.

| Prompt | | Adapter | | Train | Dataset-Specific Accuracy | | | | | | | | Universal Accuracy | |
|---|---|---|---|---|---|---|---|---|---|---|---|---|---|---|
| Sing. | Cond. | Stat. | Stoc. | Param. | CUB | Cars | SOP | InShop | NABird | Dog | Flowers | Aircraft | Unified | Harmonic |
| ✓ | ✗ | ✗ | ✗ | 0.05M | 82.8 | 51.0 | 74.7 | 72.9 | 78.3 | 85.5 | 99.2 | 50.8 | 72.3 | 70.8 |
| ✗ | ✓ | ✗ | ✗ | 0.13M | 82.8 | **54.7** | **76.8** | 76.6 | 78.7 | 85.8 | **99.3** | 53.6 | **74.0** | **73.0** |
| ✗ | ✗ | ✓ | ✗ | 2.41M | 74.5 | 81.3 | 83.7 | **90.4** | 74.5 | 81.3 | 99.0 | 66.3 | 79.4 | 80.8 |
| ✗ | ✗ | ✗ | ✓ | 2.41M | 83.6 | 83.9 | 83.8 | 89.9 | **79.2** | 84.6 | **99.4** | 71.9 | 81.1 | 83.9 |
| ✓ | ✗ | ✓ | ✗ | 2.41M | 79.6 | 80.0 | 83.8 | **90.3** | 73.7 | 81.0 | 99.0 | 65.4 | 79.3 | 80.5 |
| ✗ | ✓ | ✗ | ✓ | 2.49M | **83.9** | **84.3** | **84.0** | 89.8 | **79.2** | 84.1 | **99.3** | 72.6 | 81.3 | 84.1 |

Table 4: Recall@1 of our method compared to the best universal model (*i.e.*, CurricularFace) using different backbones and embedding dimensions. Superscripts denote embedding dimensions.

| Methods | Arch. | TotalParam. | CUB | Cars | SOP | InShop | NABird | Dog | Flowers | Aircraft | Unif. | Harm. |
|---|---|---|---|---|---|---|---|---|---|---|---|---|
| CurricularFace | ViT-S/16[128] | 21.7M | 78.3 | 77.9 | 82.0 | 89.1 | 73.0 | 79.3 | 99.1 | 65.6 | 77.9 | 79.5 |
| Ours | ViT-S/16[128] | 24.2M | **83.9** | **84.3** | **84.0** | **89.8** | **79.2** | **84.1** | **99.3** | **72.6** | **81.3** | **84.1** |
| CurricularFace | ViT-S/16[512] | 21.9M | 80.5 | 80.8 | 83.2 | 89.6 | 75.5 | 80.8 | 99.2 | 68.7 | 79.6 | 81.4 |
| Ours | ViT-S/16[512] | 24.3M | **84.6** | **85.7** | **85.1** | **90.9** | **80.1** | **84.5** | **99.3** | **74.4** | **82.3** | **85.1** |
| CurricularFace | ViT-B/16[128] | 85.9M | 79.2 | 80.4 | 84.1 | 91.3 | 75.6 | 80.3 | 99.1 | 69.2 | 80.0 | 81.5 |
| Ours | ViT-B/16[128] | 90.8M | **85.7** | **88.2** | **86.0** | **92.2** | **83.0** | **87.5** | **99.4** | **78.8** | **84.0** | **87.2** |

conditional prompt, allowing it to seamlessly accommodate varying data characteristics. Adapters, with more parameters compared to prompts, exhibit a pronounced impact on performance. In contrast to conventional static adapters, which introduce bias on larger datasets, our stochastic adapters consistently enhance results across all datasets. Notably, combining existing methods yields minimal performance gains or sometimes performance degradation, but the combination of the proposed modules boosts overall performance. Appendix B.2 shows more detailed ablation studies.

**PUMA with Different Backbones and Dimension Scales.** Table 4 presents an evaluation of performance across different embedding dimensions and backbone architectures. Given a fixed model configuration, our method consistently outperforms models fine-tuned with the state-of-the-art universal method, CurricularFace. Remarkably, our method surpasses baselines employing higher embedding dimensions or larger backbone models with over three times the number of parameters.

# 6 CONCLUSION

Previous work on deep metric learning has focused primarily on developing dataset-specific models. However, this approach is limited in terms of scalability since real-world applications usually accomoate diverse data distributions nowadays. In this paper, we thus have investigated UML, which enables a single model to manage multiple heterogeneous data distributions. In the UML setting, existing metric learning baselines suffer from imbalanced data distributions. To address this issue, we have proposed parameter-efficient tuning that is simple, lightweight, and achieving state-of-the-art performance using only a single model. We believe our work will facilitate future investigations into bridging the gap between metric learning and real-world applications.

**Ethics Statement.** In adherence to the ICLR Code of Ethics, we affirm that our research aligns with its stipulated guidelines. All datasets and pre-trained models employed in our experiments are publicly accessible and do not raise any ethical concerns.

**Reproducibility Statement.** To ensure clarity and reproducibility, we provide detailed descriptions of the two proposed modules in Sections 4.2 and 4.3. The implementation details for all baseline methods, as well as our approaches, are outlined in Section 5.1 and Appendix A. Since we propose the new settings for metric learning, called universal metric learning, we elaborate on problem formulation including details of constructing datasets and the experimental settings in Section 3.1. The source codes for experiments are available in the supplementary materials.

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

## A    IMPLEMENTATION DETAILS

We implement ours and all of the baselines in PyTorch (Paszke et al., 2017) and PyTorch Metric Learning Library (Musgrave et al., 2020b). Fair comparisons, we conduct all experiments with the following settings:

**Training.** We train models for 100 epochs using AdamW optimizer (Loshchilov & Hutter, 2019) with a weight decay of 1e-4. We use the learning rate set to 3e-5 for full fine-tuning baselines following the training setting of Hyp (Ermolov et al., 2022). We set the learning rate to 1e-4 for parameter-efficient tuning methods including ours. Following proxy anchor loss (Kim et al., 2020), we use a high learning rate for proxies in the proxy-based losses by scaling $1 \times 10^4$.

**Batch Construction.** The batch size is set to 720. For proxy-based losses, the batches are constructed by random sampling. For pair-based losses, we construct batches by first randomly sampling 180 classes, and then randomly sampling 4 images for each of the classes.

**Augmentation.** Following the standard metric learning augmentation strategy (Kim et al., 2020), training images are randomly flipped horizontally and randomly cropped with a size of 224×224. Test images are center-cropped after being resized to 256×256.

**Hyperparmeters of loss functions.** We set the hyperparameters of CurricularFace loss to a scale of $s = 32$ and a margin of $m = 0.3$. The hyperparameters for the other losses are kept at their default settings.

## B    EXTENDED ANALYSIS

### B.1    PUMA WITH DIFFERENT LOSESS

The results of utilizing different loss functions in conjunction with PUMA are summarized in Table 5. Our experiments, including the main paper, use CurricularFace as the default metric learning objective. Overall, our results demonstrate that proxy-based losses outperform pair-based losses, as shown in Table 3 in the main paper for full fine-tuning models in a UML setting. This is because smaller datasets contain fewer samples in mini-batch, making it difficult to effectively increase the distance between semantically dissimilar samples and the anchor sample. The use of Proxy-Anchor loss achieves relatively lower performance performs as it has a design that assigns samples as either positive or negative.

The use of CurricularFace loss achieves the highest performance. Losses that treat all negative samples equally during training tend to overfit specific datasets after later stages of training since the time to converge varies by dataset. In contrast, CurricularFace loss employs a design that can concentrate on hard negatives as training progresses. This allows the model to achieve high performance overall by selectively learning only difficult data in later stages, without affecting already well-learned easy data.

Table 5: Recall@1 of PUMA using different loss functions. Note that our default metric learning objective is CurriculumFace loss (Huang et al., 2020).

| Methods | Dataset-specific Accuracy | | | | | | | | Universal Accuracy | |
| --- | --- | --- | --- | --- | --- | --- | --- | --- | --- | --- |
| | CUB | Cars | SOP | In-Shop | NABirds | Dogs | Flowers | Aircraft | Unified | Harmonic |
| Triplet | 75.2 | 36.5 | 78.5 | 79.9 | 69.3 | 78.3 | 98.7 | 39.8 | 71.1 | 62.3 |
| Margin | 75.4 | 38.2 | 78.4 | 79.1 | 70.1 | 79.2 | 98.9 | 40.2 | 71.3 | 63.1 |
| MS | 72.5 | 30.8 | 80.5 | 86.1 | 66.1 | 74.9 | 98.6 | 37.2 | 71.1 | 58.9 |
| SupCon | 42.5 | 12.4 | 50.0 | 55.6 | 33.2 | 46.3 | 86.2 | 20.0 | 39.0 | 31.3 |
| PA | 82.1 | 54.8 | 83.2 | 89.6 | 77.7 | 83.4 | **99.4** | 56.1 | 77.8 | 75.2 |
| ProxyNCA++ | 49.6 | 19.1 | 62.3 | 62.3 | 39.7 | 61.3 | 93.9 | 27.1 | 52.8 | 41.3 |
| SoftTriple | 82.7 | 81.2 | 79.9 | 85.6 | 78.6 | **84.5** | 99.3 | 69.7 | 78.5 | 82.0 |
| CosFace | 83.8 | 80.9 | 82.5 | 89.3 | 78.7 | 83.8 | 99.3 | 69.0 | 80.0 | 82.6 |
| ArcFace | 82.3 | 43.6 | 80.8 | 85.9 | 77.3 | 84.0 | 99.3 | 44.9 | 75.3 | 68.8 |
| CurricularFace | **83.9** | **84.3** | **84.0** | **89.8** | **79.2** | 84.1 | 99.3 | **72.6** | **81.3** | **84.1** |

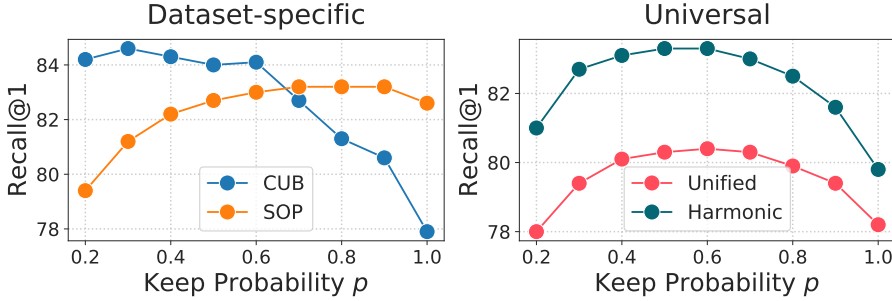

Figure 7: Recall@1 versus keep probability of adapter $p$. Note that $p = 1$ means the adapter is always used, and decreasing $p$ leads to a higher frequency of using the features of pre-trained model.

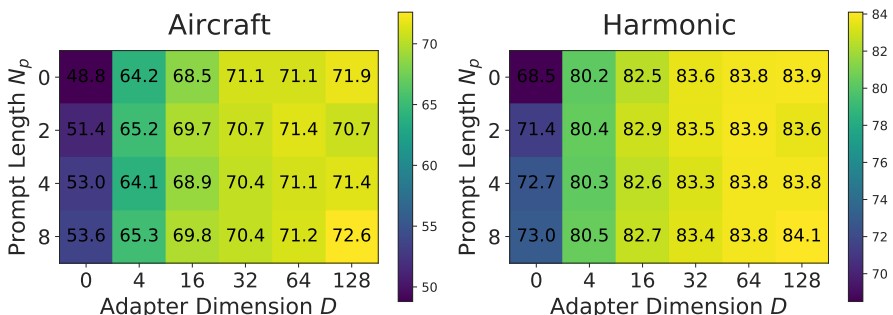

Figure 8: Ablation study on each component. The model with both prompt length and adapter dimension set to 0 indicates that only the embedding layer is being trained.

## B.2 DETAILED ABLATION STUDY

**The Effect of Stochasticity for Adapter.** Fig. 7 shows the impact of adapter stochasticity on the performance of UML. For small datasets such as CUB, decreasing $p$ leads to significant performance improvements as features from pre-trained models are frequently used instead of adapted features, mitigating bias issues. In contrast, large datasets like SOP show an opposite trend. In the end, the universal accuracy increases as $p$ decreases up to 0.5 and then decreases afterward. These results suggest that the stochasticity of the adapter plays a critical role in acquiring dataset-specific knowledge and mitigating the imbalance issue in UML.

**Ablation Study on Each Component.** Fig. 8 shows an ablation study of additional modules in PUMA. The use of the stochastic adapter and conditional prompt has a significant impact on the performance, and increasing their size gradually improves performance. The adapter has a greater impact on performance than the prompt, as the prompt uses only 0.07M parameters, which are 3% of the parameters used by the adapter. The prompt has a strong impact on smaller datasets, with a 4.8% improvement when the adapter is not used and around 1% improvement when the adapter is used.

## B.3 SCALING-UP BACKBONE

Table 6 summarizes the results of our method using a larger vision transformer backbone and its corresponding number of parameters. As the size of the pre-trained backbone increases, the overall performance improves significantly across all 8 datasets. Notably, the performance on smaller datasets shows greater improvements compared to that of larger datasets, except for the Flowers dataset, where performance is almost saturated. As a result, we observe substantial improvements in the harmonic mean accuracy metric. As the model size increases from ViT-B to ViT-L, performance improvements are observed across all datasets, but an increase in the number of parameters leads to a more pronounced bias towards large datasets. To mitigate this, attaching stochastic adapters to some layers, as in the ViT-L$^{\dagger}$ model, can be helpful.

Table 6: Recall@1 of PUMA using different backbones. The model denoted by † corresponds to a ViT-L architecture with adapters attached to the first 12 transformer layers, while the remaining models attach adapters to all transformer layers. The rest of the hyperparameters are the same.

| | Params (M) | Dataset-specific Accuracy | | | | | | | | Universal Accuracy | |
|---|---|---|---|---|---|---|---|---|---|---|---|
| Arch. | Train / Total | CUB | Cars | SOP | In-Shop | NABirds | Dogs | Flowers | Aircraft | Unified | Harmonic |
| ViT-S$^{128}$ | 2.5 / 24.2 | 83.9 | 84.3 | 84.0 | 89.8 | 79.2 | 84.1 | 99.3 | 72.6 | 81.3 | 84.1 |
| ViT-B$^{128}$ | 5.0 / 90.8 | 85.7 | 88.2 | 86.0 | 92.2 | 83.0 | 87.5 | 99.4 | 78.8 | 84.0 | 87.2 |
| ViT-L$^{128}$ | 12.9 / 316.2 | 86.6 | 88.7 | 86.9 | 93.2 | 84.4 | 90.8 | 99.4 | 78.5 | 84.8 | 88.2 |
| ViT-L$^{128†}$ | 6.6 / 309.9 | 86.8 | 90.5 | 86.3 | 92.8 | 85.3 | 90.9 | 99.3 | 79.5 | 84.9 | 88.6 |

Table 7: Performance comparison of LoRA (Hu et al., 2021) variants and their combination with PUMA. The bottleneck dimension of LoRA is set to 128, the same as that of the Adapter. The keep probability for the stochastic version of LoRA is set to $p = 0.5$.

| | Params (M) | Dataset-specific Accuracy | | | | | | | | Universal Accuracy | |
|---|---|---|---|---|---|---|---|---|---|---|---|
| Methods | Train / Total | CUB | Cars | SOP | InShop | NABirds | Dogs | Flowers | Aircraft | Unified | Harmonic |
| LoRA | 2.4 / 24.1 | 77.0 | 70.9 | 81.3 | 86.2 | 70.8 | 79.1 | 98.9 | 59.7 | 76.1 | 76.5 |
| Stochastic LoRA | 2.4 / 24.1 | 83.0 | 71.7 | 80.0 | 83.4 | 78.0 | **84.7** | **99.4** | 64.3 | 77.3 | 79.4 |
| PUMA + LoRA | 4.8 / 26.5 | 76.8 | 77.6 | 83.9 | 90.4 | 72.2 | 79.5 | 98.9 | 65.0 | 78.7 | 79.3 |
| PUMA + Stochastic LoRA | 4.8 / 26.5 | 83.1 | 83.8 | **84.7** | **91.2** | **83.9** | 73.5 | 99.2 | 72.2 | **81.5** | 83.9 |
| PUMA | 2.5 / 24.2 | **83.9** | **84.3** | 84.0 | 89.8 | 79.2 | 84.1 | 99.3 | **72.6** | 81.3 | **84.1** |

Table 8: Comparison of the performance between pair-based losses with and without cross-batch memory.

| | Dataset-specific Accuracy | | | | | | | | Universal Accuracy | |
|---|---|---|---|---|---|---|---|---|---|---|
| Methods | CUB | Cars | SOP | InShop | NABirds | Dogs | Flowers | Aircraft | Unified | Harmonic |
| Triplet | 75.2 | 36.5 | 78.5 | 79.9 | 69.3 | 78.3 | 98.7 | 39.8 | 71.1 | 62.3 |
| MS | 72.5 | 30.8 | 80.5 | 86.1 | 66.1 | 74.9 | 98.6 | 37.2 | 71.1 | 58.9 |
| Triplet + XBM | 78.4 | 50.6 | 82.8 | 89.9 | 72.8 | 79.3 | 98.9 | 51.3 | 76.0 | 71.6 |
| MS + XBM | 78.1 | 55.1 | 84.3 | 91.5 | 72.4 | 79.6 | 98.0 | 54.7 | 77.0 | 73.7 |

As the size of the backbone increases, the number of parameters in our approach also increases, but to a lesser extent relative to the backbone. Therefore, even with larger backbones, PUMA enables memory-efficient training compared to full-finetuning.

### B.4 LoRA FOR UML

LoRA (Hu et al., 2021) is one of the popular methods for parameter-efficient transfer learning in the NLP field, in addition to prefix/prompt tuning (Lester et al., 2021; Li & Liang, 2021; Wang et al., 2022; Smith et al., 2022) and adapter (Pfeiffer et al., 2020). We investigate whether LoRA effectively operates in the UML task and whether it can be used in conjunction with PUMA to boost performance. We evaluate the performance of plain LoRA, its stochastic variant, and their combination with PUMA, and summarize the results in Table 7. The results show that applying LoRA directly to UML does not work well. Like adapter, deterministic usage of LoRA leads to a significant bias towards large datasets such as SOP and In-Shop, which can be overcome by introducing stochasticity. However, despite having almost the same number of parameters as PUMA, LoRA achieves significantly lower performance. On the other hand, incorporating stochastic LoRA with PUMA results in performance improvements for the SOP, In-Shop, and NABirds datasets. However, smaller datasets exhibit decreased performance, suggesting that the two methods are not complementary and that increasing capacity through an increase in the number of parameters leads to performance improvements.

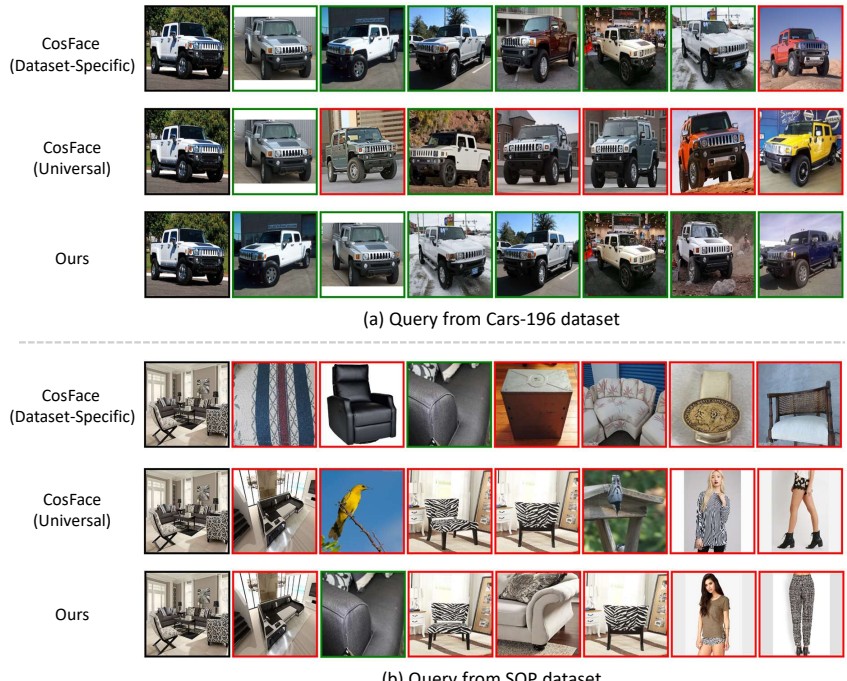

Figure 9: Qualitative results comparison of a dataset-specific model using CosFace, universal model using CosFace, and our method. For the dataset-specific and proxy anchor models, retrieval results within the same dataset are shown, while for the universal model, results using all datasets as the gallery are presented. Queries are shown in the leftmost column, followed by the top 7 retrievals. True matches and false matches are colored green and red, respectively.

## C  FURTHER DISCUSSION

### C.1  DISCUSSION ON LOSS FUNCTIONS FOR UML

**Pair-based Losses.** The experiments in our main paper and Table 5 clearly demonstrate that pair-based losses do not perform well on the UML task due to the under-sampling issue in small datasets caused by dataset imbalance. To further show that the under-sampling issue hinders the performance of pair-based losses, we present the performance of models using triplet (Schroff et al., 2015) and MS (Wang et al., 2019b) losses with cross-batch memory (*i.e.*, XBM) (Wang et al., 2020) in Table 8. While the performance overall improves when using a memory module with the two pair-based losses, it still lags behind other proxy-based losses. This is because using a memory module increases the exposure frequency of negative samples, but the dominance of large dataset samples in the memory leads to a bias towards those samples. These results suggest that completely new sampling techniques are necessary for pair-based losses to work well in the UML task. Existing techniques such as batch sampling, sample mining, and memory used in metric learning cannot address the issue of insufficient negative samples in mini-batches for UML. To address this problem, we need to explore new sampling methods that enable balanced data sampling without accessing the IDs of the datasets. Additionally, it is important to frequently supply hard negative samples to the mini-batches. This approach would be critical for achieving better performance in the UML task. While not addressed as a major issue in our main paper, addressing this under-sampling issue is quite challenging and important for future UML research.

**Proxy-based Losses.** Proxy-based losses normally achieve higher performance than pair-based losses as they introduce proxies, which reduces dependency on mini-batch sampling. However, some methods underperform compared to pair-based losses, with ProxyNCA++ (Teh et al., 2020) being a notable example. This loss modifies the softmax loss to use cosine metric and does not enforce a margin, unlike other proxy-based losses. In contrast, ArcFace (Deng et al., 2019) enforces a relatively large margin compared to other losses. CurricularFace (Huang et al., 2020) loss dynam-

Table 9: Comparison of performance between dataset-specific training and universal training. (-) and (+) denote degradation and enhancement in model performance achieved when transitioning from dataset-specific training to universal training, respectively.

| Methods | CUB | Cars | SOP | InShop | NABird | Dog | Flowers | Aircraft | Harmonic |
|---------|-----|------|-----|--------|--------|-----|---------|----------|----------|
| (a) *Models by Dataset-specific Training* | | | | | | | | | |
| PA | 80.2 | 83.7 | 84.4 | 91.5 | 69.6 | 84.2 | 99.0 | 67.9 | 81.4 |
| CurricularFace | 79.7 | 81.3 | 83.2 | 88.2 | 75.3 | 81.2 | 99.1 | 63.9 | 80.4 |
| PUMA | 81.7 | 83.9 | 85.2 | 89.9 | 77.0 | 82.5 | 99.5 | 68.9 | 82.7 |
| (b) *Models by Universal Training* | | | | | | | | | |
| PA | 77.2 (-3.0) | 73.1 (-10.6) | 83.7 **(-0.7)** | 91.9 (+0.4) | 71.5 (+1.9) | 78.1 (-6.1) | 96.4 (-2.6) | 62.7 (-5.2) | 71.0 (-10.4) |
| CurricularFace | 78.3 (-1.4) | 77.9 (-3.4) | 82.0 (-1.2) | 89.1 **(+0.9)** | 73.0 (-2.3) | 79.3 (-1.9) | 99.1 (+0.0) | 65.6 (-1.7) | 79.5 (-0.9) |
| PUMA (Ours) | **83.9 (+2.2)** | **84.3 (+0.4)** | 84.0 (-1.2) | 89.8 (-0.1) | **79.2 (+2.2)** | 84.1 (1.6) | **99.3 (-0.2)** | 72.6 (+3.7) | **84.1 (+1.4)** |

ically changes the margin based on the training progress and sample hardness, allowing it to achieve high performance. These results suggest that the margin size plays a critical role in proxy-based losses and has a significant impact on their performance. To ensure that proxy-based losses work well for the UML task, it is essential to *adaptively assign margins according to the data* and train a flexible embedding space.

## C.2 METRIC LEARNING BY DATASET-SPECIFIC VS. UNIVERSAL

Fig. 9 illustrates the image retrieval results of dataset-specific models using CosFace loss, a universal model trained using CosFace loss, and our proposed method. As shown in Fig. 9(a), the dataset-specific model retrieves relevant samples to the query as the nearest neighbors, while the universal model retrieves samples that look alike to the query but belong to different classes. Fig. 9(b) presents retrieval results using challenging query with various objects and complex patterns. The dataset-specific model retrieves one true match among its neighbors, while the rest of the results are irrelevant samples. The universal model using CosFace only retrieves samples with similar patterns from other datasets.

The presented results demonstrate that training a model on multiple datasets, despite using the same loss function, can significantly degrade the ability to distinguish classes of a specific dataset due to the negative influence of other datasets. This is a unique and challenging issue in UML research that needs to be continuously addressed in future studies. In contrast, our method enables accurate retrieval of samples belonging to the same class as the query, even though it is trained in a universal manner. These results indicate that our model can effectively recognize both dataset-specific discriminative features and common discriminative features. It is an important finding that can contribute to resolving the aforementioned issue in UML research.

## C.3 DATASET-SPECIFIC TRAINING OF PUMA

In the context of dataset-specific training of PUMA, Table 9 provides a clear insight into the performance dynamics. Notably, it showcases the substantial improvements our method achieves when compared to models trained with universal training.

The key takeaway here is that while other methods tend to become biased towards large-scale datasets when subjected to universal training, primarily learning features associated with those datasets, our method takes a different path. This remarkable boost in performance is attributed to our method's ability to effectively learn and leverage features shared across various datasets. Especially, through conditional prompt learning, our method effectively acquires an understanding of the shared characteristics among the diverse datasets. This learning process enables the model to tap into the common features present across these datasets, contributing significantly to the performance improvements observed.

Furthermore, the incorporation of the stochastic adapter proves pivotal. It helps our model address the bias that can often emerge when handling large-scale datasets during universal training. This adaptability ensures that our approach remains impartial and doesn't disproportionately favor specific datasets, ultimately resulting in more robust and balanced model performance.

In essence, these results demonstrate the ability of our method to capture common features across datasets, which results in notable enhancements in model performance. This capability sets our

Table 10: Recall@1 of baselines and ours on the four standard benchmark datasets. Superscripts denote their embedding dimensions. We note that the result of "Hyp" presented in the table is obtained from our reimplementation.

| Methods | Params (M) Train / Total | Dataset-specific CUB | Cars | SOP | InShop | Universal Unified | Harmonic |
|---|---|---|---|---|---|---|---|
| *(a) **Dataset-specific models** with CNN Backbone* | | | | | | | |
| Margin[128] (Wu et al., 2017) | 95.1 / 95.1 | 63.9 | 79.6 | 72.7 | - | - | - |
| MIC[128] (Roth et al., 2019) | 95.1 / 95.1 | 66.1 | 82.6 | 77.2 | 88.2 | - | 77.6 |
| MS[512] (Wang et al., 2019b) | 47.3 / 47.3 | 65.7 | 84.1 | 78.2 | 89.7 | - | 78.4 |
| PA[512] (Kim et al., 2020) | 47.3 / 47.3 | 68.4 | 86.1 | 79.1 | 91.5 | - | 79.2 |
| NSoftmax[512] (Zhai & Wu, 2018) | 98.2 / 98.2 | 61.3 | 84.2 | 78.2 | 86.6 | - | 76.2 |
| PNCA++[512] (Teh et al., 2020) | 98.2 / 98.2 | 69.0 | **86.5** | 80.7 | 90.4 | - | 80.7 |
| *(b) **Dataset-specific models** with ViT Backbone* | | | | | | | |
| Triplet[128] (Schroff et al., 2015) | 86.9 / 86.9 | 81.1 | 75.2 | 80.2 | 87.4 | 61.5 | 80.7 |
| Margin[128] (Wu et al., 2017) | 86.9 / 86.9 | 79.4 | 78.0 | 79.8 | 86.0 | 66.0 | 80.7 |
| MS[128] (Wang et al., 2019b) | 86.9 / 86.9 | 80.0 | 83.7 | 81.4 | 90.8 | 65.3 | 83.8 |
| PA[128] (Kim et al., 2020) | 86.9 / 86.9 | 80.2 | 83.7 | 84.4 | 91.5 | 62.7 | 84.8 |
| SoftTriple[128] (Qian et al., 2019) | 86.9 / 86.9 | 80.5 | 80.0 | 82.9 | 88.7 | 68.4 | 82.9 |
| CosFace[128] (Wang et al., 2018) | 86.9 / 86.9 | 78.8 | 83.2 | 83.2 | 89.6 | 67.2 | 83.5 |
| ArcFace[128] (Deng et al., 2019) | 86.9 / 86.9 | 76.8 | 79.4 | 83.4 | 90.3 | 67.1 | 82.2 |
| CurricularFace[128] (Huang et al., 2020) | 86.9 / 86.9 | 79.7 | 81.3 | 83.2 | 88.2 | 67.6 | 83.0 |
| Hyp[128] (Ermolov et al., 2022) | 86.9 / 86.9 | 78.8 | 78.2 | 83.6 | 91.5 | 28.3 | 82.7 |
| *(c) **Universal Models** with ViT Backbone* | | | | | | | |
| Triplet[128] (Schroff et al., 2015) | 21.7 / 21.7 | 69.5 | 35.6 | 79.8 | 85.9 | 76.0 | 60.0 |
| Margin[128] (Wu et al., 2017) | 21.7 / 21.7 | 70.0 | 37.1 | 79.8 | 82.4 | 75.7 | 60.7 |
| MS[128] (Wang et al., 2019b) | 21.7 / 21.7 | 62.5 | 23.8 | 80.2 | 87.6 | 75.0 | 48.8 |
| PA[128] (Kim et al., 2020) | 21.7 / 21.7 | 75.4 | 72.6 | 84.0 | **91.7** | 83.6 | 80.2 |
| SoftTriple[128] (Qian et al., 2019) | 21.7 / 21.7 | 76.8 | 76.9 | 82.3 | 89.2 | 82.6 | 81.0 |
| CosFace[128] (Wang et al., 2018) | 21.7 / 21.7 | 72.9 | 74.3 | 83.0 | 90.2 | 82.6 | 79.5 |
| ArcFace[128] (Deng et al., 2019) | 21.7 / 21.7 | 62.3 | 20.4 | 57.7 | 49.1 | 53.2 | 38.8 |
| CurricularFace[128] (Huang et al., 2020) | 21.7 / 21.7 | 76.9 | 77.5 | 82.7 | 89.4 | 83.0 | 81.3 |
| Hyp[128] (Ermolov et al., 2022) | 21.7 / 21.7 | 75.5 | 56.8 | 83.7 | 90.3 | 81.7 | 74.2 |
| Ours[128] | 2.5 / 24.2 | **82.5** | 84.6 | **84.9** | 91.5 | **85.7** | **85.7** |

approach apart from others, ensuring that the model doesn't fall into the bias trap associated with universal training, where it may predominantly focus on features from large-scale datasets.

# D SUPPLEMENTARY RESULTS

## D.1 RESULTS ON FOUR STANDARD BENCHMARK DATASETS

In addition to the findings highlighted in the main paper, we also extend our investigation to the realm of universal metric learning. Following the standard benchmark protocol, we employ four widely used datasets: CUB, Cars-196, Stanford Online Product (SOP), and In-Shop Clothes Retrieval (In-Shop). Our evaluation involves a comparison with state-of-the-art methods, each trained individually on these datasets. A summary of these results is provided in Table 10.

Our method outperforms existing CNN-based state-of-the-art dataset-specific models by a significant margin, even surpassing models with a 512 embedding dimension. In cases where a ViT-based backbone is employed, our approach consistently achieves higher performance compared to dataset-specific models across most datasets. While these models excel on individual datasets, their performance notably degrades when assessed on a unified dataset. Furthermore, these dataset-specific models are memory-intensive, necessitating the retention of multiple models for selection or ensemble purposes. In contrast, our universal approach handles various data distributions with a single model, eliminating the need for model selection or ensemble. Although other universal models exhibit biases towards larger datasets, PUMA is unique in maintaining its performance across both large and small datasets. This distinct capability leads to PUMA's superiority in terms of both dataset-specific accuracy and universal accuracy.

Table 11: Recall@$k$ (R@$k$) of metric learning baselines and ours with the eight datasets.

| | CUB | | | Cars | | | SOP | | | InShop | | | NABirds | | | Dogs | | | Flowers | | | Aircraft | | |
|---|---|---|---|---|---|---|---|---|---|---|---|---|---|---|---|---|---|---|---|---|---|---|---|---|
| Methods | R@1 | R@2 | R@4 | R@1 | R@2 | R@4 | R@1 | R@10 | R@100 | R@1 | R@10 | R@20 | R@1 | R@2 | R@4 | R@1 | R@2 | R@4 | R@1 | R@2 | R@4 | R@1 | R@2 | R@4 |
| (a) *Dataset-specific models by full fine-tuning* | | | | | | | | | | | | | | | | | | | | | | | | |
| Triplet | 81.1 | 88.1 | 92.9 | 75.2 | 84.2 | 90.2 | 80.2 | 84.9 | 88.6 | 87.4 | 92.1 | 95.2 | 75.2 | 83.4 | 89.5 | 81.0 | 88.4 | 93.3 | 99.1 | 99.5 | 99.7 | 64.7 | 76.6 | 86.6 |
| Margin | 79.4 | 87.8 | 92.3 | 78.0 | 86.0 | 91.9 | 79.8 | 84.6 | 88.5 | 86.0 | 91.6 | 94.9 | 74.6 | 83.1 | 89.4 | 80.3 | 87.8 | 92.7 | 99.0 | 99.5 | 99.7 | 66.8 | 79.0 | 87.1 |
| MS | 80.0 | 87.0 | 91.8 | 83.7 | 90.3 | 94.1 | 81.4 | 85.5 | 89.0 | 90.8 | 93.7 | 95.9 | 68.1 | 77.4 | 84.3 | 75.8 | 83.8 | 89.6 | 97.4 | 98.4 | 98.8 | 64.7 | 77.6 | 86.3 |
| PA | 80.2 | 87.8 | 92.4 | 83.7 | 90.2 | 94.6 | 84.4 | 88.2 | 91.1 | 91.5 | 94.5 | 96.7 | 69.6 | 78.4 | 85.5 | 84.2 | 91.1 | 94.7 | 99.0 | 99.4 | 99.6 | 67.9 | 78.1 | 87.1 |
| SoftTriple | 80.5 | 88.0 | 92.0 | 80.0 | 88.2 | 93.2 | 82.9 | 86.8 | 90.0 | 88.7 | 92.7 | 95.5 | 75.9 | 84.0 | 89.9 | 82.1 | 89.3 | 94.0 | 99.4 | 99.7 | 99.8 | 65.4 | 77.3 | 86.7 |
| CosFace | 78.8 | 86.6 | 91.5 | 83.2 | 89.5 | 93.9 | 83.2 | 87.1 | 90.1 | 89.6 | 93.4 | 95.7 | 71.4 | 80.4 | 87.0 | 79.2 | 87.1 | 92.3 | 99.2 | 99.6 | 99.7 | 61.4 | 74.3 | 83.7 |
| ArcFace | 76.8 | 85.1 | 90.9 | 79.4 | 86.5 | 91.6 | 83.4 | 87.3 | 90.1 | 90.3 | 93.7 | 96.2 | 61.0 | 70.8 | 79.1 | 76.1 | 84.2 | 90.2 | 99.2 | 99.6 | 99.7 | 60.0 | 74.1 | 84.7 |
| CurricularFace | 79.7 | 87.9 | 92.4 | 81.3 | 88.7 | 93.6 | 83.2 | 87.3 | 90.4 | 88.2 | 92.7 | 95.4 | 75.3 | 84.0 | 89.7 | 81.2 | 88.9 | 93.9 | 99.1 | 99.6 | 99.8 | 63.9 | 76.5 | 85.6 |
| Hyp | 78.8 | 87.1 | 92.3 | 78.2 | 85.6 | 91.0 | 83.6 | 88.0 | 91.1 | 91.5 | 95.2 | 97.1 | 71.0 | 79.7 | 86.4 | 72.6 | 82.0 | 89.1 | 98.7 | 99.3 | 99.5 | 65.7 | 78.2 | 86.7 |
| (b) *Universal models by full fine-tuning* | | | | | | | | | | | | | | | | | | | | | | | | |
| Triplet | 74.5 | 83.6 | 90.6 | 35.4 | 47.7 | 60.5 | 80.2 | 92.3 | 97.3 | 85.7 | 97.1 | 98.1 | 68.2 | 78.4 | 86.0 | 77.1 | 87.1 | 92.4 | 98.7 | 99.4 | 99.8 | 40.9 | 52.2 | 64.2 |
| Margin | 72.5 | 83.1 | 90.5 | 36.7 | 47.8 | 59.8 | 80.0 | 92.1 | 97.3 | 84.1 | 96.8 | 97.9 | 67.4 | 77.8 | 85.5 | 74.8 | 84.9 | 91.5 | 98.5 | 99.2 | 99.5 | 40.4 | 51.4 | 62.7 |
| MS | 66.3 | 77.1 | 85.0 | 22.9 | 32.6 | 44.7 | 78.9 | 90.9 | 96.7 | 87.2 | 96.0 | 87.2 | 58.6 | 69.4 | 78.5 | 69.8 | 80.4 | 88.0 | 97.3 | 98.4 | 98.8 | 31.5 | 42.7 | 54.5 |
| PA | 77.2 | 85.4 | 90.5 | 73.1 | 82.1 | 88.5 | 83.7 | 93.5 | 97.3 | 91.9 | 98.1 | 98.7 | 71.5 | 80.2 | 86.8 | 78.1 | 86.3 | 91.4 | 96.4 | 97.3 | 97.0 | 62.7 | 74.9 | 84.4 |
| SoftTriple | 78.9 | 87.0 | 91.7 | 77.0 | 85.8 | 91.6 | 81.3 | 92.1 | 86.6 | 88.6 | 97.4 | 98.2 | 73.8 | 82.8 | 89.2 | 79.3 | 87.6 | 92.5 | 99.1 | 99.5 | 99.8 | 64.4 | 77.6 | 86.1 |
| CosFace | 74.2 | 83.4 | 89.2 | 73.5 | 82.5 | 88.7 | 82.5 | 92.4 | 96.4 | 90.0 | 97.4 | 98.3 | 69.7 | 78.9 | 85.9 | 74.1 | 83.8 | 89.8 | 98.7 | 99.2 | 99.5 | 59.7 | 72.1 | 81.8 |
| ArcFace | 70.8 | 80.0 | 86.3 | 25.9 | 36.7 | 49.3 | 63.9 | 72.8 | 79.0 | 58.9 | 74.7 | 77.8 | 64.0 | 72.8 | 79.9 | 70.3 | 79.0 | 85.9 | 97.2 | 98.5 | 99.1 | 31.7 | 43.4 | 56.1 |
| CurricularFace | 78.3 | 87.0 | 91.7 | 77.9 | 86.4 | 92.1 | 82.0 | 92.7 | 96.9 | 89.1 | 97.5 | 98.5 | 73.0 | 82.2 | 88.7 | 79.3 | 87.7 | 92.9 | 99.1 | 99.4 | 99.6 | 65.6 | 78.0 | 99.3 |
| Hyp | 79.2 | 87.8 | 92.7 | 60.6 | 73.1 | 82.8 | 83.5 | 93.9 | 97.7 | 90.9 | 98.2 | 98.9 | 73.6 | 82.6 | 89.2 | 81.9 | 89.5 | 94.3 | 99.1 | 99.5 | 99.8 | 56.3 | 69.0 | 79.9 |
| Ours | 83.9 | 90.2 | 93.5 | 84.3 | 90.3 | 94.3 | 84.0 | 93.7 | 97.5 | 89.8 | 98.0 | 98.6 | 79.2 | 86.7 | 91.9 | 84.1 | 90.7 | 94.5 | 99.3 | 99.6 | 99.8 | 72.6 | 82.7 | 94.3 |

Table 12: Mean Average Precision at R (M@R) and R-Precision (RP) of metric learning baselines and ours with the eight datasets.

| | CUB | | Cars | | SOP | | InShop | | NABirds | | Dogs | | Flowers | | Aircraft | | Harmonic | |
|---|---|---|---|---|---|---|---|---|---|---|---|---|---|---|---|---|---|---|
| Methods | M@R | RP | M@R | RP | M@R | RP | M@R | RP | M@R | RP | M@R | RP | M@R | RP | M@R | RP | M@R | RP |
| (a) *Dataset-specific models with ViT Backbone* | | | | | | | | | | | | | | | | | | |
| PA | 46.2 | 55.5 | 27.8 | 38.3 | **51.5** | **54.2** | 65.5 | 68.1 | 37.0 | 47.1 | 48.5 | 58.8 | 91.7 | **93.3** | 21.4 | **34.7** | 40.5 | 51.6 |
| CosFace | 43.4 | 53.8 | 26.4 | 36.8 | 49.6 | 52.5 | 63.6 | 66.3 | 35.9 | 46.1 | 47.3 | 57.7 | **92.1** | 93.0 | 20.1 | 32.8 | 38.8 | 49.9 |
| CurricularFace | 43.8 | 53.5 | 24.0 | 34.8 | 48.9 | 52.2 | 62.0 | 65.3 | 36.4 | 46.8 | 48.4 | 58.8 | 91.1 | 93.0 | 17.9 | 31.3 | 37.0 | 49.1 |
| (b) *Universal models by full fine-tuning* | | | | | | | | | | | | | | | | | | |
| PA | 41.9 | 51.5 | 22.9 | 34.1 | 50.7 | 53.7 | **67.7** | **70.3** | 33.0 | 43.4 | 39.5 | 51.0 | 70.3 | 74.0 | 18.5 | 31.8 | 35.4 | 47.3 |
| CosFace | 36.6 | 46.7 | 21.3 | 32.1 | 48.4 | 51.4 | 64.1 | 66.8 | 28.7 | 39.2 | 30.4 | 42.7 | 85.1 | 87.0 | 17.3 | 30.1 | 32.3 | 44.3 |
| CurricularFace | 40.8 | 50.7 | 22.3 | 33.4 | 47.6 | 50.8 | 62.8 | 65.7 | 31.6 | 42.6 | 38.3 | 50.2 | 87.2 | 88.8 | 17.6 | 30.5 | 34.3 | 46.5 |
| Ours | **48.5** | **57.5** | **28.1** | **38.5** | 50.3 | 53.3 | 65.2 | 68.0 | **38.8** | **49.1** | **48.7** | **59.0** | 91.4 | 92.5 | **21.8** | 34.4 | **41.1** | **52.0** |

Remarkably, our proposed method achieves exceptional results. It outperforms the best universal model by 3.1% in unified accuracy and 4.4% in harmonic mean accuracy. Notably, it even surpasses the state-of-the-art dataset-specific models by 17.3% in unified accuracy and 0.9% in harmonic average accuracy. This remarkable performance is achieved without employing hyperparameters tailored to individual datasets.

## D.2 RESULTS IN OTHER METRICS

The table 11 presents the Recall@$k$ (R@$k$) of metric learning baselines and our method on the eight datasets. We report R@$k$ for the SOP and In-shop datasets following their convention, and we report the results of R@1, R@2, and R@4 for the remaining datasets. The results show that our method achieves significantly better performance than the existing methods in terms of R@$k$ overall.

We also adopt the enhanced metric, Mean average precision at R (M@R) and R-Precision (RP), proposed in Musgrave et al. (2020a) to thoroughly evaluate both our model and the baseline approaches. Table 12 shows the comparative analysis between our method and the three top-performing baseline models. Interestingly, our findings indicate that even when using the same loss function, universal models demonstrate a more pronounced performance degradation in M@R and RP on smaller datasets compared to what is observed when employing the Recall@$k$ metric. Remarkably, our method consistently outperforms not only the universal models but also the dataset-specific models across diverse datasets, mirroring the previous results. This achievement finally leads to the

attainment of higher harmonic mean accuracy. Significantly, this highlights that the superiority of our method is not confined solely to the Recall@$k$ metric, but rather is grounded in its capacity to establish a well-clustered embedding space across all datasets.

### D.3 $t$-SNE VISUALIZATION

Fig. 10 shows $t$-SNE visualizations of universal embedding space learned by our method on multiple datasets. The left side shows the visualization of the training set, while the right side shows that of the unseen test set. The visualization of the training set reveals that each dataset and each class is well-clustered. Similarly, the test set visualization also shows well-clustered groups for each dataset and class, with CUB and NABirds, both bird species datasets, having semantically similar data points clustered together. In conclusion, our results demonstrate that nearest neighbor data points in the embedding space are semantically similar, suggesting that our model learns universal semantic similarity across multiple datasets.

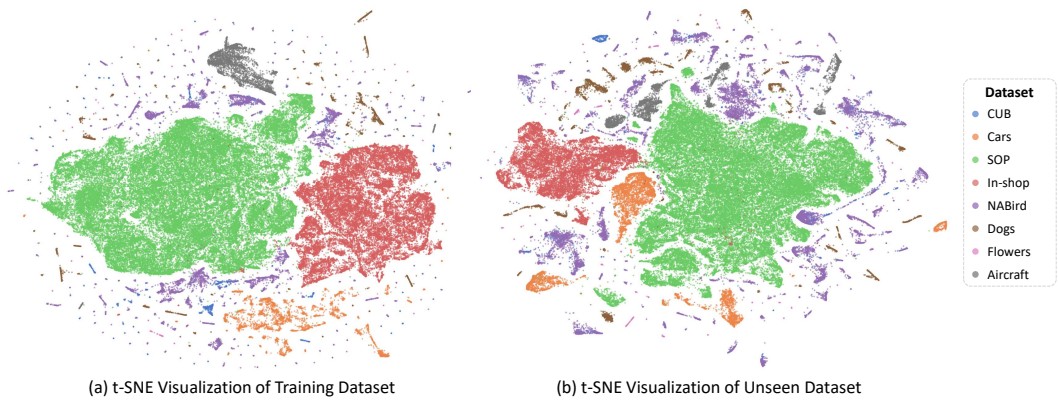

(a) t-SNE Visualization of Training Dataset      (b) t-SNE Visualization of Unseen Dataset

Figure 10: $t$-SNE visualization of universal embedding space.

