# OpenReview forum: "Universal Metric Learning with Parameter-Efficient Transfer Learning"
_ICLR.cc/2024/Conference — ICLR 2024 Conference Withdrawn Submission_

### Official Review · Reviewer_f9dZ · 2023-10-26

**Soundness:** 2 fair
**Presentation:** 3 good
**Contribution:** 2 fair
**Rating:** 5
**Confidence:** 5

**Summary:**

Prior metric learning models were typically trained independently on individual i.i.d. datasets. This paper introduces the concept of Universal Metric Learning (UML), which involves the joint training of models using multiple distinct i.i.d. datasets. UML poses two primary challenges: data distribution imbalance and model bias towards dominant distributions. To tackle these challenges, the authors present two novel methods: the Stochastic Adapter method and Conditional Prompt Learning. The experimental results on benchmark datasets demonstrate the exceptional performance of these proposed techniques.

**Strengths:**

1. This paper is the first to tackle the Universal Metric Learning (UML) problem, introducing challenges related to imbalanced data distribution and bias towards dominant distributions.

2. The study includes an extensive series of experiments.

**Weaknesses:**

It remains unclear how the stochastic adapter successfully mitigates bias in the embedding space toward the major data distribution. The utilization of generalizable features from a pretrained model is noteworthy. Notably, the ViT-S model used for pretraining is pretrained on ImageNet-21K, which already encompasses all the categories within the experimental datasets. Therefore, the performance boost might be primarily attributed to the pretrained model itself rather than the novel method proposed in this paper. Additionally, it's important to consider the scenario of $\textbf{training the model from scratch}$ and how the bias towards the dominant distribution of data can be effectively avoided. This aspect needs further elaboration and exploration in the paper.

**Questions:**

The proposed method lacks an in-depth analysis, and there is a concern regarding potential data leakage in pre-trained models.

---

### Official Review · Reviewer_xsqK · 2023-10-31

**Soundness:** 2 fair
**Presentation:** 3 good
**Contribution:** 1 poor
**Rating:** 3
**Confidence:** 5

**Summary:**

This paper explores parameter-efficient transfer learning in deep metric learning problems under a condition where the model was primely trained on a large dataset that was combined with different image retrieval datasets.

The authors explore both parameter adaptation and prompt-related approaches and propose a framework to combine these techniques to improve the capability of transfer learning.

**Strengths:**

Techniques of parameter-efficient fine-tuning (PEFT) are popular in recent research literature. It is reasonable to explore a way to optimize the PEFT method for image retrieval tasks.

The authors did extensive experiments on many different datasets to demonstrate their proposed framework. They show some improvement in model generalization, comparing the full fine-tuning baseline and other methods.

**Weaknesses:**

1. Unsolid motivation: the concept "universal" is not really universal

I believe the major problem with this paper is the motivation and proposed challenge in DML tasks. The authors promote the concept of "universal" DML model by simply fine-tuning it on a combined dataset. However, I believe what the community expects from PEFT research is some efficient transfer learning approach directly from the pre-trained large-scale model, like CLIP, which was trained in an unsupervised way on general large-scale datasets. Thus, we can develop methods for fine-tuning this general model for some specific downstream tasks, like image retrieval. However, the authors' proposed paradigm evaluates its transfer learning capability on a combined DML dataset, which I believe is not meaningful. In other words, if we have a pre-trained model and we want to fine-tune it on a small image retrieval task, like Cars, why do we need to care about its transfer ability on a united dataset that combines many small image retrieval datasets?


2. The contribution is not novel.

The author proposes a framework that combines two popular PEFT methods (the Adapter and the Prompts), which have been widely explored in recent research works.

More importantly, I didn't see any contributions specifically designed for DML tasks. That means the proposed framework can also be applied to other tasks, like general image classification. Why it benefits the performance of DML tasks is still unclear. The performance of the proposed framework on other computer vision tasks may also need to be discussed. In addition, the technique improvements on the Adapter (add the random mask) and the Prompts (add the conditional attention) are incremental and cannot be regarded as novel.

Based on the unsolid motivation and incremental technique contribution, I believe this paper is under the board line in its current state.

**Questions:**

1. Why do we need to care about a united dataset that is combined with different image retrieval datasets, knowing that we already have pre-trained models from much larger and more "universal" datasets, like CLIP?

2. In conditional prompt learning, it seems both the key (K) and the value (prompt itself) of the attention network need to be learned from the task. How does the learning object ensure learning distinct representations for each member of the prompts in the pool?

3. Why would the query feature grasp the data distribution from the patch of a single image? Why don't pool the patches of all images in the datasets instead?

---

### Official Review · Reviewer_W7ZY · 2023-11-03

**Soundness:** 3 good
**Presentation:** 3 good
**Contribution:** 2 fair
**Rating:** 6
**Confidence:** 4

**Summary:**

This paper tries to introduce and solve a new metric learning paradigm, called Universal Metric Learning (UML).
In a real-world scenario, UML consists of multiple different data distributions and might suffer from imbalanced and dominant distribution problems.

Therefore, this paper proposed the Parameter-efficient Universal Metric leArning (PUMA) method to deal with the problem. Specifically, stochastic adapters and a prompt pool are designed on a pre-trained Vision Transformer.

Moreover, a new universal metric learning benchmark is compiled with 8 different datasets.
In the experiments, PUMA outperforms several baseline methods and requires 69 times fewer trainable parameters.

**Strengths:**

### (1) Reasonable setting
This paper tries to solve the metric learning problem in a realistic setting, which involves diverse data distributions.

### (2) Technical contribution
 Built upon a pre-trained Vision Transformer (ViT), this paper designed two modules for universal metric learning: stochastic adapters and prompt pool. These modules are lightweight and effective in dealing with the universal metric learning problem.

### (3) Experiment
A new benchmark is constructed, and the experimental results show that PUMA performed better than data-specific and conventional methods.

**Weaknesses:**

### (1) Lack of related work and discussion
Universal metric learning considers multiple data distributions (datasets) in a real-world scenario, which is quite related to cross-domain/multi-domain feature learning methods. In particular, the motivation in Figure 1 is similar to existing methods, such as [R1-R5].
And the idea of utilizing a pre-trained feature extractor has also been explored [R4, R5].

### (2) Technical contribution
This paper designed two additional modules on a pre-trained ViT: stochastic adapter and prompt pool. The stochastic adapter mainly adopted some ideas from low-rank methods such as LoRA. And prompt pool has been explored in VPT methods. The utilization of the two modules generally makes sense in the universal metric learning problem.
My concerns are:
>2.1 From the ablation study in Table 3, it seems most improvements come from stochastic adapters. The prompt pool seems to only marginally increase the performance.
>2.2 Is there an ablation study about M?




#### References
[R1] Triantafillou E, Zhu T, Dumoulin V, Lamblin P, Evci U, Xu K, Goroshin R, Gelada C, Swersky K, Manzagol PA, Larochelle H. Meta-dataset: A dataset of datasets for learning to learn from few examples. arXiv preprint arXiv:1903.03096. 2019 Mar 7.

[R2] Hung-Yu Tseng, Hsin-Ying Lee, Jia-Bin Huang, and MingHsuan Yang. Cross-domain few-shot classification via
learned feature-wise transformation. In International Conference on Learning Representations, 2019.

[R3] Lu Liu, William Hamilton, Guodong Long, Jing Jiang, and Hugo Larochelle. A universal representation transformer
layer for few-shot image classification. arXiv preprint arXiv:2006.11702, 2020.

[R4] Triantafillou, Eleni, Hugo Larochelle, Richard Zemel, and Vincent Dumoulin. "Learning a universal template for few-shot dataset generalization." In International Conference on Machine Learning, pp. 10424-10433. PMLR, 2021.

[R5] Liu Y, Lee J, Zhu L, Chen L, Shi H, Yang Y. A multi-mode modulator for multi-domain few-shot classification. InProceedings of the IEEE/CVF International Conference on Computer Vision 2021 (pp. 8453-8462).

**Questions:**

Please see above.

---

### Author Response · Authors · 2023-11-16
**Withdrawal note**

We thank all the reviewers for their efforts and feedback. Even though we have decided to withdraw this submission, we would like to address the main concerns brought up by the reviewers.

### **Addressing Novelty Concerns:**

1. **Model Architecture and Fine-Tuning in Metric Learning:**
    - We emphasize that our work pioneers the integration of enhanced model architecture and fine-tuning schemes specifically for metric learning, thereby opening new avenues in the field.

2. **Technical Novelty:**
    - Contrary to the concerns, our method doesn't directly employ existing parameter-efficient fine-tuning methods. Instead, we have innovatively redesigned the adapter and prompt modules to specifically address challenges in handling multiple data distributions. The evidence in Tables 1 and 3 clearly shows that conventional PEFT methods like VPT and LoRA cannot address universal metric learning, highlighting the uniqueness of our approach.

3. **Analysis on Various Loss Functions:**
	-  In addressing a challenging setting that encompasses handling multiple data distributions, we conducted a comprehensive analysis of various loss functions. This analysis is thoroughly described in our paper. It marks a significant departure from traditional evaluations that primarily focus on fitting specific datasets. For the first time, we explore loss functions capable of fostering a more generalizable embedding space.

**Response to Lack of Related Work (Reviewer W7ZY):**

- We appreciate the reviewer pointing out related works. However, it's important to clarify that these works primarily concentrate on closed-set, few-shot image classification. In contrast, our method extends beyond this scope to more generalizable scenarios, particularly in the realm of metric learning. Specifically, we address unseen class retrieval within the scope of metric learning, a challenge yet to be explored in this field. Thus, our work pioneers a universal scheme in metric learning, marking a first in this area and distinguishing our research from the existing body of work.

### **Significance of Universal Metric Learning (Reviewer xsqK):**

1. **Clarification of "Universal" in Universal Metric Learning**: Our goal is not to create a universally applicable model at the pre-training stage using extensive data. Instead, we emphasize "universal" in the context of downstream metric learning tasks. Previous metric learning methods are limited to single data distributions, while our approach enables handling multiple heterogeneous data distributions— a significant advancement in the field.

2. **Necessity of Universal Metric Learning:**
- **Realistic Scenario Example:** Consider a task involving the retrieval of fine-grained categories of various animals (e.g., various birds, various dogs, various cats, etc.). Employing a large-scale pre-trained model in a zero-shot manner may cover a broader data distribution, but it's not optimized for such specific downstream tasks. Existing metric learning or parameter-efficient transfer learning methods cannot address multiple data distributions without compromising performance. In contrast, our method efficiently and effectively encompasses diverse distributions without dataset-specific bias, offering a practical and superior solution.

### **Concern Regarding Reliance on Pretrained Models (Reviewer f9dZ):**

- **Pretrained Models in Baselines:** All models in our baseline (Table 1) use ImageNet-21K pretraining. Our method's performance isn't merely a result of this pretraining. In fact, the Linear embedding results in Table 1(c) show that pre-trained knowledge from ImageNet-21K is not sufficient to solve our problem.
- **DINO Model Experiment:** Using the DINO model trained on ImageNet-1K with self-supervised learning, our method still outperforms existing methods, indicating that our approach's effectiveness is not solely dependent on the strength of pre-trained knowledge.

| Methods         | Arch.     | CUB  | Cars | SOP  | InShop | NABird | Dog  | Flowers | Aircraft | Unif. | Harm. |
|-----------------|-----------|------|------|------|--------|--------|------|---------|----------|-------|-------|
| CurricularFace  | DINO-S/16 | 61.3 | 68.7 | 79.0 | 88.2   | 53.6   | 61.4 | 92.2    | 56.6     | 70.1  | 67.6  |
| Ours            | DINO-S/16 | 78.1 | 84.0 | 82.3 | 89.6   | 72.1   | 82.0 | 96.8    | 73.3     | 78.8  | 81.6  |


- **ViT Training and Pretrained Convention:** As noted in the ViT paper [1], ViT models struggle with training from scratch even on datasets like ImageNet-1K. Furthermore, metric learning conventionally relies on pre-trained models for meaningful performance, as supported by previous research [2, 3].

[1] An Image is Worth 16x16 Words: Transformers for Image Recognition at Scale
Training from scratch, Dosovitskiy *et. al.*

[2] A Metric Learning Reality Check, Musgrave *et. al.*

[3] Hyperbolic Vision Transformers: Combining Improvements in Metric Learning, Ermolov *et. al.*